# Risankizumab: Efficacy, Safety, and Survival in the Mid-Term (52 Weeks) in Real Clinical Practice in Andalusia, Spain, According to the Therapeutic Goals of the Spanish Psoriatic Guidelines

**DOI:** 10.3390/life12111883

**Published:** 2022-11-14

**Authors:** Ricardo Ruiz-Villaverde, Lourdes Rodriguez-Fernandez-Freire, Amalia Pérez-Gil, Pilar Font-Ugalde, Manuel Galán-Gutiérrez

**Affiliations:** 1Dermatology Department, Hospital Universitario San Cecilio, 18016 Granada, Spain; 2Instituto Biosanitario de Granada (Ibs), 18012 Granada, Spain; 3Dermatology Department, Hospital Universitario Virgen del Rocio, 41013 Sevilla, Spain; 4Dermatology Department, Hospital Universitario Virgen de Valme, 41014 Sevilla, Spain; 5Rheumatology Department, Hospital Universitario Reina Sofía, 14004 Córdoba, Spain; 6Dermatology Department, Hospital Universitario Reina Sofía, IMIBIC, 14004 Córdoba, Spain

**Keywords:** risankizumab, psoriasis, long-term, real-world evidence

## Abstract

Introduction. Risankizumab is a humanized monoclonal antibody of the immunoglobulin G1 (IgG1) type that binds selectively, and with high affinity, to the p19 subunit of interleukin-23 (IL-23), resulting in the inhibition of inflammation and clinical symptoms associated with psoriasis. Its introduction has managed to increase the levels of efficacy and safety (improving upon those previously presented by the anti-IL-23 class). Material and methods. Retrospective analysis of a multicenter, observational study of real clinical practice, including patients with moderate-to-severe plaque psoriasis in treatment with risankizumab. This cross-sectional analysis includes information on patients from May 2020 to June 2022. A total of six tertiary hospitals in Andalusia (Spain) participated in this study. Analyses were performed “as observed” using GraphPad Prism version 8.3.0 for Windows. Results. Regarding the percentage of patients who reached PASI 90 or PASI 100 at week 52, 92.5% achieved the therapeutic goal of PASI 90, and 78.5% reached PASI 100. When analyzing the results by absolute PASI, we found that 78.5% (*n* = 33) obtained PASI 0, 85.7% (*n* = 36) obtained PASI ≤ 1, and all patients achieved PASI ≤ 3 (disease control). Discussion. Risankizumab has shown promising results in the control of psoriasis in the long-term, with a high percentage of patients (>80%) maintaining PASI 90 and PASI 100 up to 52 weeks of treatment. No abnormal safety findings have been reported, and risankizumab appears to be a solid treatment in the different scenarios analyzed.

## 1. Introduction

Psoriasis (PSO) in a multisystem chronic inflammatory disease with a significant impact on the quality of life in patients who suffer from it. The latest epidemiological study carried out in Spain estimates its prevalence at 2.3% [1] in the biological era after a first approximation in previous years, where the prevalence is similar in both sexes, and it is estimated to be 1.17–1.43%. The highest prevalence rates were shown among 20- to 50-year-old subjects [2].

The IL-23 inhibitors guselkumab (GUS), tildrakizumab (TIL), and risankizumab (RIS) represent the latest class of biologics to be approved for the management of moderate-to-severe psoriasis [3]. Its incorporation confirms the importance of the IL-17/IL-23 axis in moderate-to-severe psoriasis from the pathophysiological point of view. Of the three drugs, both guselkumab and risankizumab have recently been approved by the EMA and the FDA for the treatment of psoriatic arthritis.

Risankizumab is a selective, humanized immunoglobulin G1 (IgG1) monoclonal antibody directed against interleukin (IL)-23 protein. The efficacy and safety of risankizumab have been evaluated in 2.109 patients with moderate-to-severe plaque psoriasis as candidates for systemic treatment in four pivotal phase III, multicenter, randomized, double-blind, placebo-controlled studies: UltIMMa-1 [4], UltIMMa-2 [4], and IMMhance [5]; with active treatment using ustekinumab in the UltIMMa-1 and UltIMMa-2 studies; and with adalimumab in the IMMvent study [6]. In all four studies, the criteria for primary outcomes were: an improvement of at least 90% on the Psoriasis Intensity and Severity Index (PASI 90) and a Physician’s Global Assessment (sPGA) score of “clear” or “nearly clear” (sPGA 0/1) at week 16. The trials included a 16-week period of controlled treatment (induction); a maintenance period continuing until week 52 in the UltIMMa-1 and -2 studies, or through week 44 in the IMMvent study; and a long-term, open-label, extension phase in the M15-997 (LIMMITLESS) study [6]. Furthermore, the IMMhance study, after the initial double-blind period of 28 weeks, investigated the effect of the withdrawal and reintroduction of RIS.

Our research team previously published the results of our patient series to verify the evaluation of RIS over a short-term period (16 weeks) with promising results [7]. 

## 2. Objective

The main objective of this study is to evaluate the efficiency and safety of risankizumab (RIS) in the medium-term (52 weeks) in patients with moderate-to-severe psoriasis (PSO) and to compare the results with data previously published in other real clinical practice series and previous clinical trials.

## 3. Material and Methods

### 3.1. Study Design

Retrospective analysis of a multicenter, observational study of real clinical practice, including patients with moderate-to-severe plaque psoriasis in treatment with risankizumab (RIS). In this cross-sectional analysis, the information for patients from May 2020 to June 2022 has been included. A total of six tertiary hospitals in Andalusia (Spain) participated in this study. This study has been approved by the Ethics Committee of the Hospital Universitario San Cecilio (DER-HUSC-2022-008). All patients provided valid informed consent for the use of their anonymized data in academic and research processes.

### 3.2. Patients

Inclusion criteria were as follow: (1) adult (>18 years old) moderate-to-severe plaque PSO patients. Also included were patients with an initial diagnosis of plaque psoriasis who developed erythrodermic or pustular psoriasis at some point in their evolution, with this even being the reason for starting treatment with RIS as an approved off-label medication. A PASI >10 was considered to be a moderate-to-severe PASI. No differentiation was made between a moderate and a severe PASI since there is no uniformity of classification criteria among the different dermatological societies; (2) PSO diagnosis ≥ 1 year; (3) patients who experienced primary or secondary failure or unspecified inefficacy with another systemic or biological treatment, as well as patients who had discontinued treatment due to adverse events (AEs); and (4) patients on risankizumab treatment of 150 mg s.c. (at weeks 0 and 4, followed by a maintenance dose every 12 weeks, according to the summary of product characteristics). In at least two hospitals, patients received training in subcutaneous injection techniques so as to self-inject the medication. 

Exclusion criteria were as follow: (1) types of PSO other than psoriasis vulgaris, with the previously mentioned exceptions; (2) an inability to sign the informed consent document.

### 3.3. Outcome Measures

#### 3.3.1. Efficacy

Disease severity and treatment response was assessed using the absolute psoriasis area and severity index (PASI), body surface area (BSA), VAS pruritus, and the Dermatology Life Quality Index (DLQI). Primary failure was considered as the inability to reach PASI 90 or PASI < 3 within 16 weeks of treatment, and secondary failure was defined as the inability to maintain PASI 90 or PASI < 3 after 16 weeks of treatment if it had been previously achieved. These cut-off points have been evaluated according to the updated recommendations for the management of moderate-to-severe psoriasis from the Psoriasis Group (GEP) of the Spanish Academy of Dermatology and Venereology (AEDV) [8]. As this is a retrospective study of real clinical practice, the *n* of patients may differ according to the cut-off point considered for the exploitation of results.

#### 3.3.2. Safety

The safety and tolerability to risankizumab were evaluated during the follow-up portion of the study. (Any adverse event experienced by the patient was reported.) The analysis included discontinuations due to a lack of effectiveness and for safety reasons 

Routine blood chemistry tests were performed at the follow-up visits at the clinicians’ discretion. 

### 3.4. Statistical Analysis

A descriptive analysis of the evaluated variables was carried out. Means and standard deviations were calculated for quantitative variables, and absolute values and percentages were calculated for categorical variables. Means were compared using Student’s t-test. To address differences between groups, the Chi [2] test or Fisher’s exact test was used to analyze categorical variables, and Mann–Whitney U test was performed to analyze continuous variables; a *p* value < 0.05 was considered statistically significant. Analyses were performed “as observed” using GraphPad Prism version 8.3.0 for Windows (GraphPad Software, San Diego, CA, USA, www.graphpad.com (accessed on 15 September 2022)). 

This study was conducted in accordance with the Helsinki Declaration of 1964 and all subsequent amendments. 

## 4. Results

Our study included 78 patients, 49 men (62.8%) and 29 women (37.2%), with moderate-to-severe psoriasis treated with RIS who had completed at least 12 weeks of treatment, while 42 patients reached week 52. At the time of the analysis, all of the included patients continued in treatment, and therefore, there were no suspensions, which made calculating survival using the corresponding Kaplan–Meier curve unfeasible.

The baseline characteristics of the sample are shown in Table 1. The mean age of our patients was 51.6 years, and the mean BMI of our patients was 29.83 kg/m^2^. We also designated two subgroups based on BMI, with 80% of patients having a value equal to or greater than 25 kg/m^2^ (overweight and obesity).

The most frequent comorbidities in our patients were dyslipidemia and hypertension (both at 32%). It is important to emphasize that 19 patients had an active smoking habit (24.3%), 15 had psoriatic arthropathy (19.2%), and 11 patients had associated NAFLD (14.1%). In our cohort, it is noteworthy that there were 8 patients with solid neoplasia (4 patients with colorectal carcinoma, 3 patients with breast cancer, and 1 patient with Hodgkin’s lymphoma, all of them in remission) prior to the start of treatment, which was not considered a contraindication for treatment after consensus was reached with the oncology services of the respective hospitals. All of these patients reached at least PASI 90 in their respective endpoints according to the cut made for data evaluation.

In addition, 13 patients with latent tuberculosis infection were included, with 1 of them not undergoing prophylactic treatment, without evidence of the reactivation of tuberculosis infection. There were also 2 HIV positive patients who did not develop complications.

In relation to the therapies previously used, 17 patients (21.8%) were naïve to biologics, with the largest group of patients being those who had previously received a biologic treatment (34.6%). Seven patients had previously used five or more biological treatments, reflecting the therapeutic complexity of some of these patients. Table 1 shows the percentage of patients who had received various lines of biological treatment. Interestingly, regarding the last drug prescribed, we observed that the largest group comprised patients who had failed another drug of the anti-IL-23 family (*n* = 26). The last treatment that had been used did not influence the response, with that being excellent, regardless of the previous drug exposure.

In terms of efficacy, our patients had mean baseline pruritus scores of 5.94, mean PASI scores of 12.48, mean BSA scores of 17.56, and mean DLQI scores of 15. By week 16, all had significantly decreased, with readings of 1.58 PASI and 1.79 BSA. At week 28, these mean values for PASI and BSA were 1.02 and 1.58, respectively.

According to the effect of gender on response, we noted that women had better efficacy results, which may be considered as a predictor of a good response. This fact may be related to the lower mean of comorbidities compared to that of the male gender (mean of comorbidities: 1.8 vs. 2.1). However, in the bivariate analysis, we did not find statistically significant differences. We also investigated whether the presence of psoriatic arthropathy had any effect on the therapeutic response, noting that the patients who suffered from it had better responses according to the efficacy parameters (PASI and BSA), as well as a decrease in the PROs score (VAS pruritus and DLQI). Considering whether BMI (dividing the patients into two groups with the limit established at 25 kg/m^2^), had any influence on the efficacy parameters allowed us to verify that, although the response was good in both groups, a better response was seen in the subgroup with a BMI < 25 kg. /m^2^, which performed as expected in daily clinical practice. In relation to survival, we observed that all of the patients maintained the treatment without presenting notable adverse effects.

Regarding the percentage of patients who reached PASI 90 or PASI 100 at week 52, 92.5% achieved the therapeutic goal of PASI 90, and 78.5% reached PASI 100. When analyzing the results by absolute PASI, 78.5% (*n* = 33) obtained PASI 0, 85.7% (*n* = 36) obtained PASI ≤ 1, and all patients achieved PASI ≤ 3 (disease control) (Figure 1).

Figure 2 shows how the clinically relevant and optimal therapeutic objective was achieved at each cut-off point according to the sample size achieved. Finally, and as a point of interest, we correlated the PROs measured in patients who reached week 52 (DLQI 0/1) and VAS pruritus = 0 with the efficacy outcomes to see how their response was better when the therapeutic response was optimal (Figure 3 and Figure 4).

## 5. Discussion

The inclusion and exclusion criteria in randomized clinical trials give a theoretical measure of drug efficacy under optimal conditions. These conditions are often distinct from the scenarios that are observed in real clinical practice (patients with comorbidities, i.e., metabolic syndrome and previous solid or hematological neoplasia) and also usually have a washout period, which is ignored in clinical practice where the overlapping of treatments can occur. For this reason, real-life studies represent an opportunity and an important source of data in the decision-making process when choosing the best treatment for our patients; also, it incorporates their voices in the measurement of patient reported outcomes (PROs) (i.e., DLQI and VAS pruritus).

In Table 2, we have collected the main series of real clinical practice studies that have evaluated the efficacy and safety of RIS in the long-term (>40 weeks), where PASI 90 and PASI 100, or equivalent measures, have been collected as measures of efficacy. From the point of view of the effectiveness of the treatment, our series presents a percentage of patients who reached PASI 90 (PASI < 1) equivalent to that of other series in real clinical practice although the percentage of patients who reached PASI 100 (PASI = 0) is higher, without reaching the data presented by Caldarola [9].

Gkalpakiotis et al. [10] present, what is to date, one of the largest series of patients treated with RIS at 52 weeks (*n* = 154) with the same baseline characteristics observed in the rest of the published series. Absolute PASI values were also analyzed in this study, and at week 52 (*n* = 34 patients), 79.4% and 88.2% reached PASI < 1 and PASI < 3, respectively. No new safety signals were observed, and only one patient developed a colorectal carcinoma, which was pending therapeutic decision in collaboration with the oncologist. In our series, 8 patients had a solid neoplasia under control with an evolution time of fewer than five years, and without reactivation after the introduction of risankizumab. In their series (*n* = 166), Mastorino et al. [13] evaluated clinical-response predictor factors and found that smoking habits, joint involvement, obesity, and previous biological experience may negatively affect treatment response, while the involvement of difficult body sites had a minor impact. In bionaïve patients, the response seems to be faster than in bioexperienced patients. 

Of all the published series, we would like to make special reference to the study by Hansel et al. [17]. In this study, a breakdown of the data was made according to four main characteristics: (a) Sex (with a better response in female patients), as in our series; (b) baseline PASI > or < 20 (with very close efficacy results); (c) BMI > or < 25 (with the difference that only 53.1% of patients reached PASI 100, compared to 69.6% of patients who were not overweight who reached it; and (d) Bionaïve vs. bioexperienced patients. Although a bivariate analysis of the four scenarios explored was not performed, the differences are only more marked when the *n* of patients is lower. This issue shows of the robustness of RIS in all of them, and our results show a high consonance in this sense.

In contrast with the Spanish practice guidelines [8]. where an absolute PASI < 3 is considered a response within the therapeutic objective, the Italian guidelines (EuroGuiDerm) [18] are slightly more selective, and the Gargiulo study [16] incorporates the measurement of absolute PASI < 2 as a measure of effectiveness. Curiously, when using this measurement, the efficacy curves overlap with PASI 75 up to week 104. However, in the subanalysis performed according to different epidemiological characteristics (BMI, bionaïvity, etc.), the effectiveness results were intermediate, between PASI 75 and PASI 90, according to the data considered. We would like to emphasize that it this the only real clinical practice series that shows the results at 104 weeks (*n* = 26), results that not only maintain but improve the PASI 75, 90, and 100 data (96.2%, 80.8%, and 69.2%) compared with week 52. They did not observe differential results with respect to the UltIMMa-1 and UltIMMa-2 clinical trials, similar to our own series. The subanalysis by groups shows a worse response in patients with a BMI < 25, unlike the results of Hansel et al. [12]. It is well known that patients with a high BMI have a worse response to some biologics, such as anti-TNF, ustekinumab, and secukinumab, while with other anti-IL-23 and anti-IL-17 agents, the response seems to be more solid and independent of weight, as we discussed in our first published study [19].

Ruggiero et al. [20] present an indirect comparative study that evaluated the efficacy of guselkumab (GUS) and risankizumab (RIS) at 44 weeks in real clinical practice. Although the sample size was small (*n* = 21), 66.6% of the patients reached PASI 90, and 42.8% reached PASI 100. Especially interesting is the safety data, which show that 4.8% of patients experience diarrhea as an AE with the administration of RIS.

Although it was not the objective of our study, we believe that it is also important to highlight studies that have evaluated the efficacy of RIS in special locations, such as the genital area. In their cohort of 20 patients treated with RIS, Sotiriu et al. [21] demonstrated non-inferiority in their response as measured by sPGA-G, with a sustained decline in efficacy through week 24, and a more pronounced decline in DLQI through week 16. Regarding another special location, which can sometimes be disabling for the patient, such as nail psoriasis, the results reported by Megna et al. [14] are especially promising as Nail Psoriasis Severity Index (NAPSI) scores improved as well, becoming statistically significant only at week 16 and continuing thereafter [9.3 ± 4.7 at baseline, 4.1 ± 2.4 (*p* < 0.01) at week 16, and 1.4 ± 0.8 (*p* < 0.0001) at week 52]. 

In the series published by Metha et al. [15], two important data stand out: five patients had to intensify their treatment every 8 weeks due to persistent psoriasis (*n* = 2) or difficult-to-control psoriatic arthritis (*n* = 3), and despite the fact that no described safety findings were different from those published in clinical trials, a 46-year-old patient developed breast cancer although she had been received all conventional systemic treatments and six biologicals prior to treatment with RIS, so it is difficult to attribute its development to RIS.

To conclude, we would like to add that our experience in the management of patients with moderate-to-severe psoriasis with the other two molecules of its class (guselkumab [22] and tildrakizumab [23]) yields equivalent efficacy and safety results, so that, in a scenario of high pharmacoeconomic impact such as that in which we currently live, the prescription of one or another treatment is also influenced by the cost of acquiring them.

## 6. Limitations of the Study

Some limitations of this study include (1) the retrospective nature of the study, (2) the small sample size used, (3) the fact that safety evaluation was suboptimal due to the nature of the study (real clinical practice), and (4) the short period of the drug use. 

## 7. Conclusions

Risankizumab is a drug of the anti-IL-23 family that shows a high effectiveness profile in real clinical practice, showing its robustness in different scenarios (obesity, bionaïvity, gender, and comorbidities) and without safety findings beyond those previously reported in the pivotal clinical trial.

## Figures and Tables

**Figure 1 life-12-01883-f001:**
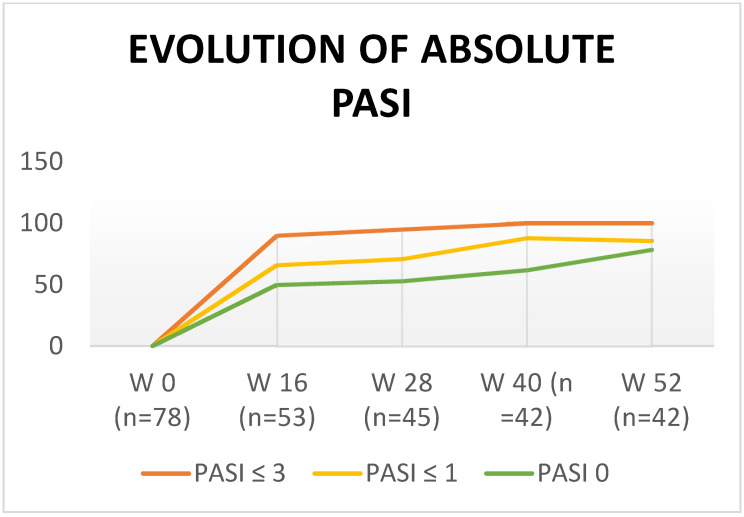
Evolution of absolute PASI.

**Figure 2 life-12-01883-f002:**
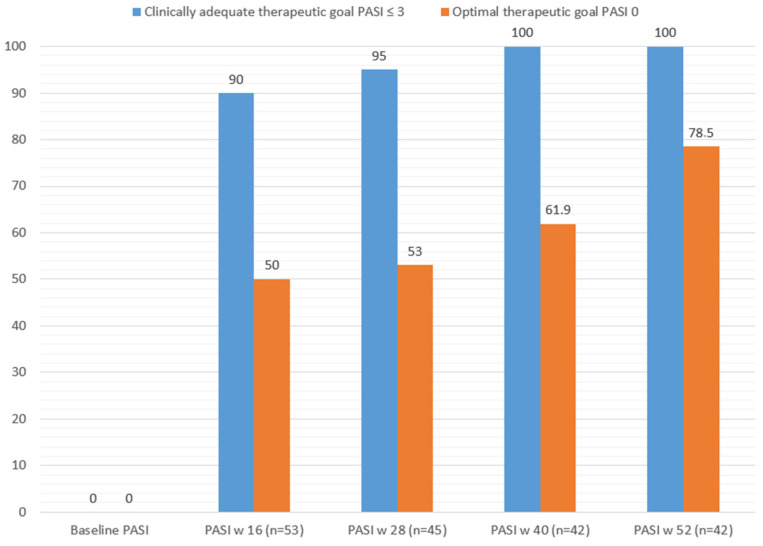
Clinical and optimal therapeutic response at every cut-off point.

**Figure 3 life-12-01883-f003:**
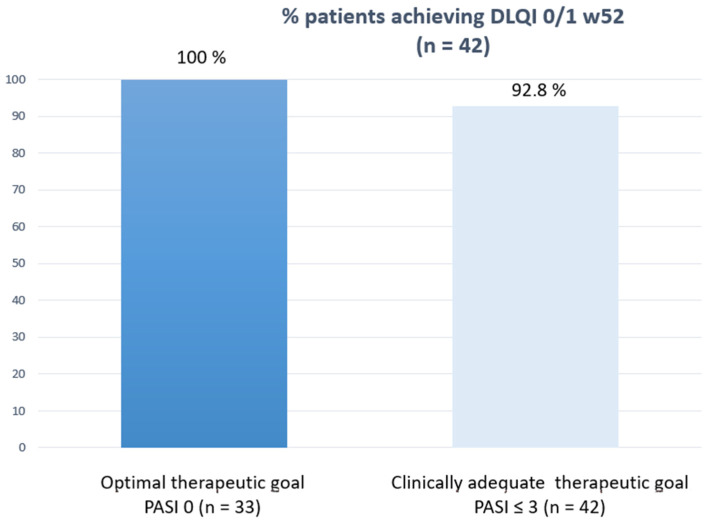
Correlation between DLQI 0/1 at w52 and efficacy outcomes.

**Figure 4 life-12-01883-f004:**
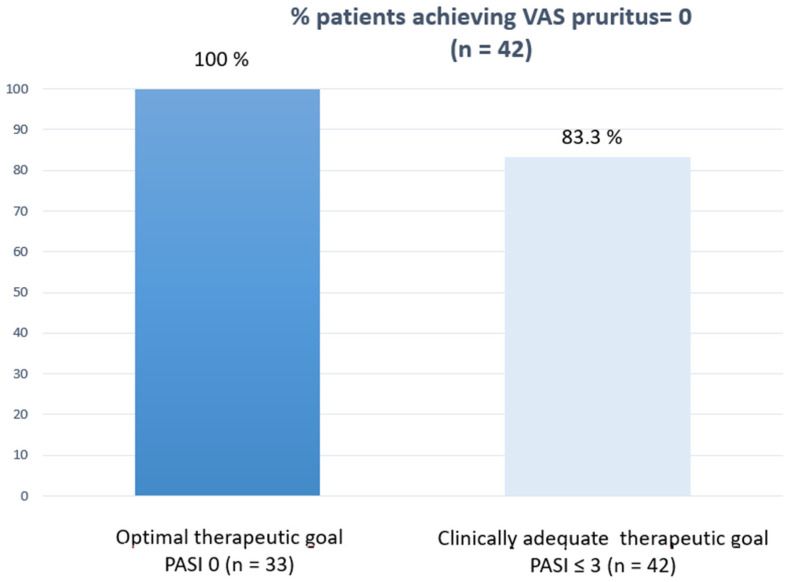
Correlation between VAS pruritus = 0 at w52 and efficacy outcomes.

**Table 1 life-12-01883-t001:** Demographic and clinical characteristics of the study population.

Basal Data	SD	*n* = 78
Age (years), media (SD)	14.84	51.6
Sex, *n* (%)		
Male		49 (62.8%)
Women		29 (37.2%)
BMI	6.30	29.83
<25 kg/m^2^		20%
≥25 kg/m^2^		80%
Psoriasis type, *n* (%)		
Plaque		74 (94.9%)
Erythroderma		2 (2.6%)
Pustulous		2 (2.6%)
Special locations, *n* (%)		
Palmoplantar psoriasis		7 (9%)
Scalp psoriasis		17 (21.8%)
Nail psoriasis		21 (26.9%)
Genital psoriasis		2 (2.6%)
Comorbidities		
Psoriatic arthritis		15 (19.2%)
Diabetes		8 (10.2%)
Smoking habit		19 (24.3%)
Hypertension		25 (32%)
Dyslipidemia		25 (32%)
Depression		8 (10.2%)
Nonalcoholic fatty liver disease (NAFLD)		11 (14.1%)
Previous solid neoplasia		8 (10.2%)
PASI w0, mean	6.92	12.48
BSA w0, mean	12	17.56
VAS pruritus w0, mean	2.95	5.94
DLQI w0, mean	6.63	15.09
Number of previous biological treatments		
0		17 (21.8%)
1		27 (34.6%)
2		17 (21.8%)
3		8 (10.3%)
4		2 (2.6%)
≥ 5		7 (8.9%)
Previous target treatment		
Anti-TNF		17 (21.8%)
Anti-IL-17		15 (19.2%)
Anti-IL-23		26 (33.3%)
Others		20 (25.6%)

BMI, body mass index; BSA, body surface area; DLQI, Dermatology Life Quality Index; PASI, psoriasis area and severity index; VAS, visual analog scale.

**Table 2 life-12-01883-t002:** Data from real-life risankizumab (long-term follow-up) studies.

Reference	Follow-Up (Weeks)	Number of Patients	PASI 90	PASI 100
Gkalpakiotis, S. [10]	52	154	82.4%	67.6%
Borroni, R.G. [11]	40	66	85.7%	62.3%
Hansel, K. [12]	52	57	85.5%	60%
Mastorino, L. [13]	52	166	82%	73%
Megna, M. [14]	52	39	84.6%	64.1%
Caldarola, G. [9]	52	63	95.24%	90.5%
Metha, M. [15]	44	70		46% (IGA = 0)
Gargiulo, L. [16]	52	131	61.1%	78.6%

## Data Availability

Data are available upon reasonable request to the corresponding author.

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
