# Peer review of "Risankizumab: Efficacy, Safety, and Survival in the Mid-Term (52 Weeks) in Real Clinical Practice in Andalusia, Spain, According to the Therapeutic Goals of the Spanish Psoriatic Guidelines"

_life, 2022, doi:10.3390/life12111883_

Round 1
Reviewer 1 Report
The analysis and description of the efficiency and safety of risankizumab in the medium term compared with data previously published it’s quite interesting. It is important understanding the high efficacy profile in real clinical practice of the therapies even for the impact on health systems especially in a pandemic era. The retrospective observational study was well done and the results are quite interesting, but there are some questions that should be clarified.
1. To better understand the gender response reported, how many women had comorbidities?
2. What previous solid neoplasia did patients have and oh many of them reached PASI 90 or 100?
3. There are few grammar mistakes. Examples of these errors are present in the following sentences: pag. 2 line 83”…Chharacteristic…”
4. Check the Institutional Review Board and Data Availability Statement
Author Response
The analysis and description of the efficiency and safety of risankizumab in the medium term compared with data previously published it’s quite interesting. It is important understanding the high efficacy profile in real clinical practice of the therapies even for the impact on health systems especially in a pandemic era. The retrospective observational study was well done and the results are quite interesting, but there are some questions that should be clarified.
AUTHOR REPLY: We thank the reviewer for their kind comments that will help improve the quality of the manuscript.
1.To better understand the gender response reported, how many women had comorbidities?
AUTHOR REPLY. This fact may be related to the lower mean of comorbidities compared to the male gender (Mean of comorbidities 1.8 vs 2.1), although in the bivariate analysis we did not find statistically significant differences)
2.What previous solid neoplasia did patients have and oh many of them reached PASI 90 or 100?
AUTHOR REPLY: All data have been uptaded. All of them reached at least PASI90 in their respective endpoints according to the cut made for data evaluation.
3.There are few grammar mistakes. Examples of these errors are present in the following sentences: pag. 2 line 83”…Chharacteristic…”
AUTHOR REPLY: It has been reviewed carefully4. Check the Institutional Review Board and Data Availability Statement
4. Check the Institutional Review Board and Data Availability Statement
AUTHOR REPLY: It has been reviewed carefully
Reviewer 2 Report
Abstract must not show abbreviations
The abbreviation RIS must be mentioned from the start, not in line 241
The abbreviation PSO must be mentioned in full frist.
Table 1 is not well arranged, and the top row does not match with all data
why did you include erythrodermic and pustular psoriasis while you mentioned before in methods you include plaque type only
what was the range of PASI and on what base you judged what is moderate and what is severe
HIV and TB patients were better not included as it was a risk
You did not mention why 25 patients did not complete the 52 weeks you must mention in detail
Author Response
We thank the reviewer for their kind comments that will help improve the quality of the manuscript.
- Abstract must not show abbreviations. AUTHOR REPLY: All abbreviations have been removed from the abstract
- The abbreviation RIS must be mentioned from the start, not in line 241. The abbreviation PSO must be mentioned in full frist. AUTHOR REPLY: It has been revised properly
- Table 1 is not well arranged, and the top row does not match with all data. AUTHOR REPLY: We have structured the table with appropriate borders to make the columns more easily identifiable.
- Why did you include erythrodermic and pustular psoriasis while you mentioned before in methods you include plaque type only. AUTHOR REPLY: We clarify this point in the material and methods section.
- What was the range of PASI and on what base you judged what is moderate and what is severe. AUTHOR REPLY: We clarify this point in the material and methods section.
- HIV and TB patients were better not included as it was a risk. AUTHOR REPLY: All patients who met the inclusion criteria according to the actual clinical practice of the different centers that participated in the study have been included.
- You did not mention why 25 patients did not complete the 52 weeks you must mention in detail. AUTHOR REPLY: As this is a retrospective study of real clinical practice, the n of patients may differ according to the cut-off point considered for the exploitation of results.
Regards
Reviewer 3 Report
Dear Author,
1. The study is interesting, however, some clarifications are required
2. The inclusion criteria was chronic plaque psoriasis but erythrodermic and pustular psoriasis were included.
3. Words like basal and media in the table in respect to PASI and DLQI are not understood.
4. Editing of english language is required
5. It is important to compare its efficacy in biologic naive and biologic experienced patients also apart from other subgroups.
Author Response
Dear Sir/Mdm
1. The study is interesting, however, some clarifications are required
AUTHOR REPLY: We thank the reviewer for their kind comments that will help improve the quality of the manuscript.
2. The inclusion criteria were chronic plaque psoriasis but erythrodermic and pustular psoriasis were included.
AUTHOR REPLY: We clarify this point in the material and methods section.
3. Words like basal and media in the table with respect to PASI and DLQI are not understood.
AUTHOR REPLY: The table has been clarified and the words have been replaced for a better understanding of it
4.Editing of English language is required
AUTHOR REPLY: English grammar has been reviewed by a native speaker
5.It is important to compare its efficacy in biologic naive and biologic experienced patients apart from other subgroups.
AUTHOR REPLY: Thank you for your comment, but this sub-analysis will be the subject of a later publication with a larger sample size.
Regards
Reviewer 4 Report
Dear authors,
A well-designed study that can contribute to the literature.
1) Remove repetitive sentences in the abstract and introduction.
2) You can read this study as evidence that IL inhibitors have a broader safety profile. ''Ataseven A, et al. Comparison of anti-TNF and IL-inhibitors treatments in patients with psoriasis in terms of response to routine laboratory parameter dynamics. Journal of Dermatological Treatment, 2022; 33(2), 1091-1096.''
3) In the Limitations section, the shortness of the drug use periods can also be added.
Best regards...
Author Response
We thank the reviewer for their kind comments that will help improve the quality of the manuscript.
A well-designed study that can contribute to the literature.
1.Remove repetitive sentences in the abstract and introduction. AUTHOR REPLY: Abstract and Introduction have been reviewed in order to remove this sentences.
2) You can read this study as evidence that IL inhibitors have a broader safety profile. ''Ataseven A, et al. Comparison of anti-TNF and IL-inhibitors treatments in patients with psoriasis in terms of response to routine laboratory parameter dynamics. Journal of Dermatological Treatment, 2022; 33(2), 1091-1096.''
AUTHOR REPLY: Thank you very much for the note of this very interesting article, but since we have not focused on the biochemical findings or comparison with TNF blockers, we do not consider its inclusion in the discussion. We will delve into its study in the manuscript that is currently in preparation.
3) In the Limitations section, the shortness of the drug use periods can also be added.
AUTHOR REPLY: It has been added
Regards